# Protease-Sensitive and -Resistant Forms of Human and Murine Alpha-Synucleins in Distinct Brain Regions of Transgenic Mice (M83) Expressing the Human Mutated A53T Protein

**DOI:** 10.3390/biom13121788

**Published:** 2023-12-13

**Authors:** Dominique Bétemps, Jean-Noël Arsac, Simon Nicot, Dominique Canal, Habiba Tlili, Maxime Belondrade, Eric Morignat, Jérémy Verchère, Damien Gaillard, Lilian Bruyère-Ostells, Charly Mayran, Latifa Lakhdar, Daisy Bougard, Thierry Baron

**Affiliations:** 1ANSES (French Agency for Food, Environmental and Occupational Health & Safety), University of Lyon, 69364 Lyon, France; dominique.betemps@anses.fr (D.B.); jean-noel.arsac@anses.fr (J.-N.A.); dominique.canal@anses.fr (D.C.); habiba.tlili@anses.fr (H.T.); eric.morignat@anses.fr (E.M.); jeremy.verchere@anses.fr (J.V.); damien.gaillard@anses.fr (D.G.); latifa.lakhdar@anses.fr (L.L.); 2Pathogenesis and Control of Chronic and Emerging Infections, University of Montpellier, Inserm, Etablissement Français Du Sang, 34493 Montpellier, France; simon.nicot@efs.sante.fr (S.N.); maxime.belondrade@efs.sante.fr (M.B.); lilian.bruyere-ostells@efs.sante.fr (L.B.-O.); charly.mayran@efs.sante.fr (C.M.); daisy.bougard@efs.sante.fr (D.B.)

**Keywords:** α-synuclein, β-amyloid, Parkinson’s disease, Lewy Body Disease, PMCA

## Abstract

Human neurodegenerative diseases associated with the misfolding of the alpha-synuclein (aS) protein (synucleinopathies) are similar to prion diseases to the extent that lesions are spread by similar molecular mechanisms. In a transgenic mouse model (M83) overexpressing a mutated (A53T) form of human aS, we had previously found that Protein Misfolding Cyclic Amplification (PMCA) triggered the aggregation of aS, which is associated with a high resistance to the proteinase K (PK) digestion of both human and murine aS, a major hallmark of the disease-associated prion protein. In addition, PMCA was also able to trigger the aggregation of murine aS in C57Bl/6 mouse brains after seeding with sick M83 mouse brains. Here, we show that intracerebral inoculations of M83 mice with C57Bl/6-PMCA samples strikingly shortens the incubation period before the typical paralysis that develops in this transgenic model, demonstrating the pathogenicity of PMCA-aggregated murine aS. In the hind brain regions of these sick M83 mice containing lesions with an accumulation of aS phosphorylated at serine 129, aS also showed a high PK resistance in the N-terminal part of the protein. In contrast to M83 mice, old APPxM83 mice co-expressing human mutated amyloid precursor and presenilin 1 proteins were seen to have an aggregation of aS, especially in the cerebral cortex, hippocampus and striatum, which also contained the highest load of aS phosphorylated at serine 129. This was proven by three techniques: a Western blot analysis of PK-resistant aS; an ELISA detection of aS aggregates; or the identification of aggregates of aS using immunohistochemical analyses of cytoplasmic/neuritic aS deposits. The results obtained with the D37A6 antibody suggest a higher involvement of murine aS in APPxM83 mice than in M83 mice. Our study used novel tools for the molecular study of synucleinopathies, which highlight similarities with the molecular mechanisms involved in prion diseases.

## 1. Introduction

Synucleinopathies in humans—including Parkinson’s disease (PD), dementia with Lewy bodies (DLB) and multiple system atrophy (MSA)—are essentially characterized by the misfolding/aggregation of the alpha-synuclein (aS) protein within intracellular lesions in neurons (PD and DLB) and/or glial cells (MSA). Most cases of these diseases are considered as sporadic, but the aggregation of aS can also be fostered by mutations or the multiplication of the SNCA encoding gene, resulting in familial cases of PD or DLB. Setting aside alpha-synuclein, which was by far the most enriched protein in PD and MSA extracts, a recent study showed that both diseases share a striking overlap of their sarkosyl-insoluble proteomes, the vast majority of which are made up of mitochondrial and neuronal synaptic proteins [1]. Whereas human neurodegenerative diseases often involve the aggregation of several proteins, Alzheimer’s disease (AD) neuropathology is common in synucleinopathies and confers a worse prognosis [2]. Mixed pathologies may in fact represent the rule rather than the exception, and may even bring into question the nosology of neurodegenerative diseases [3].

A number of recent studies have suggested that molecular mechanisms similar to those occurring in prion diseases may be involved in the progression of lesions during these diseases [4,5]. Among these studies, we reported the first observations in a transgenic mouse line (M83) [6] overexpressing the A53T mutated human aS involved in some genetic PD cases. We found that intracerebral inoculations of brain extracts from old and sick M83 mice drastically accelerated the onset of paralysis that typically develops during the ageing of these mice [7]. These observations were then confirmed in a number of other studies, either after intracerebral inoculations of fibrillar recombinant aS [8,9,10] or of brain extracts from human patients, notably with MSA [11,12,13]. Other studies used different peripheral inoculation routes [14,15,16,17] with similar findings.

More recently, we decided to explore the comparison with prions further by trying to amplify the misfolding of aS using PMCA in vitro with brain homogenate as a normal aS substrate [18]. Using repeated cycles of the sonication–incubation of a normal brain substrate seeded with minute quantities of a prion-infected sample, PMCA did indeed replicate the misfolding of the pathological prion protein, amplifying infectivity and allowing the ultra-sensitive detection of misfolded prion proteins from animals or patients with prion diseases [19,20,21,22,23,24]. Using a similar approach, we found that the PMCA of M83 mouse brain homogenate induced the aggregation of aS, which was associated with the PMCA products’ pathogenicity once inoculated into M83 mice brains. This finding was similar to that for the brain extracts from sick M83 mice containing abundant aS phosphorylated at serine 129 (pSer129-aS) [18]. PMCA also induced the aggregation of aS in C57Bl/6 mouse brains after being seeded with brain homogenate from sick M83 mice [18], raising the possibility that a transmission barrier between the human aggregated aS from the M83 mouse brain and the normal aS from the C57Bl/6 brain could have been crossed during the in vitro process. The possible contribution of murine aS during the disease remains poorly understood in M83 mice, which express both human and murine aS [8,9,25]. With few exceptions [12,26], this is generally the case in transgenic aS models, unlike the transgenic mice on a *prn*p ^0/0^ background used in most prion studies [27,28]. In wild-type mice, intracerebral inoculations of recombinant murine aS fibrils have induced a murine aS pathology, which is associated with the seeding capacity in these mice [29]. However, intra-cerebral inoculations of human aS derived from the Lewy pathology only induced a sparsely dispersed aS pathology in wild-type mice [30,31].

Resistance to proteinase K (PK) digestion is the essential marker of the pathological prion protein in both natural and experimental prion diseases. It is also the key biochemical feature that reveals that a prion protein from a normal brain substrate has been misfolded during PMCA after its seeding with a prion-infected sample containing the PK-resistant prion protein PrP^res^ [19]. The presence of C-terminally truncated aS species has been reported in M83 mice with a pathology [32,33]. In addition, our previous study of aS PMCA in the M83 mouse model revealed that PMCA-aggregated aS became PK-resistant (aS^res^), and that the aggregation of aS could be specifically recognized using ELISA methods [18].

In this new study, we further describe how biochemical features specifically found in M83-PMCA samples containing de novo aggregated aS can help to decipher the molecular pathogenesis in M83 transgenic mice expressing the human mutated A53T protein, including the compared behaviors of human and murine aS. Importantly, we herein report the pathogenicity for M83 mice of murine aS aggregated in vitro using PMCA. In addition, to further address the crucial questions associated with mixed pathologies, we describe the neuropathological features of a novel transgenic mouse model (APPxM83) [34] obtained by breeding M83 mice with an AD mouse model co-expressing a human mutated amyloid precursor protein and presenilin 1 [35].

## 2. Materials and Methods

### 2.1. Ethics Statement

Animal experiments were performed in our approved ANSES facilities (No. D69 387 0801), in accordance with EEC Directive 86/609/EEC and French decree No. 2013-118. The experimental protocols were authorized (Nos. 16-039, 19-019 and 19-042) by the ANSES/ENVA/UPEC ethic committee, ComEth.

### 2.2. Brain Samples

For PMCA experiments, brain homogenates were prepared by homogenizing whole brains using a mechanical homogenizer (grinding balls, Precellys 24, Bertin Technologies, Montigny le Bretonneux, France) to obtain 10% homogenates (*w*/*v*) in a converting buffer (phosphate buffered saline (PBS) pH = 7.2, 150 mM NaCl, 1% Triton X-100 and a cocktail of protease inhibitors). They were prepared from 2-month-old (i) transgenic mice, overexpressing either the human A53T mutated aS under the prion promoter (M83 line) [6] or the wild-type human aS under the PDGF-β promoter (D line) [36], and (ii) C57Bl/6 mice expressing (C57Bl/6) or not (C57Bl/6-KO) [37] mouse aS. C57Bl/6 and C57Bl/6-KO were purchased from Janvier labs and Envigo, respectively.

For the biochemical analysis of M83 mouse brains, aS extractions were performed as previously described [9,18] from dissected brain or spinal cord homogenates obtained in High Salt Buffer (HSB) (50 mM Tris–HCl pH 7.5, 750 mM NaCl, 5 mM EDTA, 1 mM DTT, 1% phosphatase and a cocktail of protease inhibitors). Protein measurements were performed using a DC Protein Assay (Biorad, Mitry-Mory, France).

### 2.3. Protein Misfolding Cyclic Amplification (PMCA)

The PMCA procedure has been previously described [18]. Briefly, 100 µL of 10% brain homogenates from 2-month-old mice (M83^+/−^ or ^+/+^, D^+/+^, C57Bl/6 or C57Bl/6-KO) were used directly in PMCA reactions in comparison with shaking (1000 rpm at 37 °C) or standing conditions at 37 °C. C57Bl/6 brain homogenates were also used as a substrate after mixing with brain homogenates (10^−5^ dilution) from sick M83^+/+^ mice used as pathological seeds. PMCA reactions were performed in PCR tubes containing three Teflon beads and were submitted to amplification in a microplate sonicator (Q700; Qsonica, Newton, CT, USA,). Each cycle was composed of an incubation step (29 min 40 s at 37 °C) and a sonication step (20 s at 140 W) and 1 or 2 rounds of 144 cycles were performed according to the experiment. For serial PMCA, the amplified material was diluted 1:10 in a fresh PMCA substrate and submitted to a new PMCA round. To avoid cross-contamination, experiments were carried out under strict quality-controlled PCR conditions.

### 2.4. Bioassays

Homozygous M83^+/+^ mice are described as developing characteristic motor symptoms typically between 8 and 16 months of life, beginning with reduced ambulation, balance disorders, partial paralysis of a hind limb, then progressing to prostration, difficulty in feeding, weight loss, a hunched back and general paralysis [6,16].

APPxM83 mice co-expressing A53T human aS and amyloid precursor/presenilin 1 proteins were obtained by crossing M83^+/+^ mice with APPPS1-21 mice, kindly provided by Mathias Jucker, University Medical Center Hamburg-Eppendorf, Germany [35]. The genotyping was performed with selected forward (GAATTCCGACATGACTCAGG) and reverse (GTTCTGCTGCATCTTGGACA) primers. Crossing these two transgenic lines generated M83 hemizygous ^(+/−)^ mice with (*n* = 6) or without (*n* = 7) human mutated amyloid precursor protein/presenilin 1 transgenes. Mice were followed up to the age of 22 months at which time they were sacrificed for neuropathological studies [34].

Bioassays in M83^+/+^ mice were performed by stereotactic injections unilaterally into the striatum. The injections contained 2 µL of inocula made of M83-PMCA or C57Bl/6-PMCA (M83^+/+^-seeded) samples obtained after 144 PMCA cycles with 1 or 2 rounds, respectively (Experiments 2 and 3 in Table 1), or of M83-standing samples prepared from M83^+/+^ mouse brains maintained in standing conditions (at 37 °C without shaking) during the same time period. For this last condition, a first experiment (Experiment 1 in Table 1) has been previously described [18] in which mice were sacrificed at the age of 14 months. This was after the end of a parallel experiment with an M83-PMCA sample, though two mice (11–12 months old) died either with signs of paralysis or from intercurrent disease. We performed a second experiment in which a randomly chosen mouse was sacrificed as a control whenever one mouse in the inoculated group showed the typical clinical signs of paralysis (Experiment 2 in Table 1, inoculation with the M83-PMCA sample). Two final control experiments were performed with stereotactic injections of samples made of C57Bl/6-KO brain spiked with a 10^−5^ dilution of sick M83^+/+^ mouse brain homogenate (without PMCA) or of unseeded C57Bl/6-KO brain treated by two PMCA rounds (Experiments 4 and 5, respectively, in Table 1). The inoculated M83^+/+^ mice were housed in groups in enriched cages in a temperature-controlled room on a 12h light/dark cycle and received water and food ad libitum. The mice were monitored daily to detect any symptoms of the M83 disease as described [7,16], including slow ambulation, balance disorders and partial paralysis of a hind limb. When mice developed these first symptoms or any intercurrent disease not associated with the typical M83 disease phenotype, they were sacrificed after intracardiac perfusion of Tris buffered saline (TBS) under anesthesia by an intraperitoneal injection of xylazine (10 mg/kg) and ketamine (50 mg/kg).

Finally, for the analysis of the aggregation of aS during the ageing of M83 mice using an ELISA method, an experiment was performed in which a cohort of M83^+/+^ mice was monitored and six mice were sacrificed at each different timepoint (aged 10 days, then 1, 2, 3.5, 5, 8 and 10 months). Brains and spinal cords were dissected and analyzed using 4D6/C-20R and Syn303/D37A6 ELISAs.

### 2.5. Western Blot

PMCA samples or brain homogenates were mixed with 4× Laemmli denaturing buffer (Biorad, Mitry-Mory, France) in order to load ~10 µg proteins per lane. Proteins were separated after heat denaturation (5 min at 100 °C) on TGX FastCast acrylamide gels (Biorad, Mitry-Mory, France) loaded with denatured PMCA samples or brain homogenates, then blotted onto PVDF membranes (Immobilon-P, Millipore-Sigma, Burlington, MA, USA). The aS was cross-linked to the membranes using 4% paraformaldehyde and 0.01% glutaraldehyde diluted in PBS for 30 min. Unspecific binding sites were saturated by adding 5% milk for 1 h. Proteins were detected by either C-20R (1:10,000; Santa Cruz Biotechnology, Dallas, TX, USA), clone 42 (1/4000; BD Biosciences, Franklin Lakes, NJ, USA), D37A6 (1:1000; Cell Signaling technology, Danvers, MA, USA), 4B12 (1:5000; Biolegend, Amsterdam, Netherlands), EP1536Y (1:4000; Abcam, Cambridge, UK (ab51253)) MJFR1 (1/5000, Abcam), Syn303 (1:2000; Biolegend) and 4D6 (1:4000, Abcam) aS antibodies diluted in PBS-Tween 20 0.05% (PBST) overnight at +4 °C (Appendix A [6,32,38,39,40,41,42,43,44]). Membranes were then incubated with anti-mouse or anti-rabbit HRP (1:10,000; Clinisciences, Nanterre, France), for 1 h at room temperature (RT). The signals were then revealed with a chemiluminescent substrate (Supersignal WestDura, ThermoFisher Scientific, Waltham, MA, USA) and analyzed using the ChemiDoc system and ImageLab software (version number 5.2.1) (Bio-Rad, Mitry-Mory, France).

For some experiments, we performed proteinase K (PK) digestion on PMCA samples or brain homogenates. The PK concentration ranged from 1 to 1000 µg/mL (final concentration) and the temperature was 37 °C for 30 min. Prior to migration onto 12% TGX FastCast acrylamide gels (Biorad), 10 µg of proteins from digested samples was mixed with 4× Laemmli denaturing buffer.

### 2.6. ELISA

Alpha-synuclein was detected from 10% (*w*/*v*) mouse brain homogenates in HSB using an ELISA, as previously described [9,16,18]. Briefly, plates (MaxiSorp^TM^, Thermo Scientific Nunc, Waltham, MA, USA) were coated with capture antibodies, either 4D6 (1:4000) or Syn303 (1:1000) (Biolegend, Amsterdam, The Netherlands) antibodies in 50 mM Na_2_CO_3_/NaHCO_3_ (pH9.6) with 100 µL per well at 4 °C overnight. Plates were washed five times in PBS with 0.05% Tween 20 (PBST). Superblock T20 PBS blocking buffer (Pierce) was then added for 1 h at RT with shaking at 150 rpm. The plates were again washed five times in PBST and 20 µg proteins from the brain homogenates (dilution dependent on protein dosage by DC Protein Assay (Biorad) in PBST BSA 1%) were incubated at 25 °C for 2 h with shaking. After washing five times with PBST, captured aS was detected by different anti-aS antibodies (Appendix A [6,32,38,39,40,41,42,43,44]).

To analyze the PMCA samples, aS was detected in some experiments with 4D6 or C-20R antibodies only, without a capture antibody, or with the C-20R antibody after using the 4D6 capture antibody. The plates were washed five times in PBST and either anti-mouse (Southern Biotech, Birmingham, AL, USA) or anti-rabbit (Southern Biotech) IgG HRP conjugates were added at 1:2000 dilution in PBST BSA 1% for 1 h at RT. After washing the plates five times with PBST, 100 µL of 3,3′,5,5′- tetramethylbenzidine (TMB) solution (Sigma-Aldrich, Saint Louis, MO, USA) was added to each well and incubated for 15 min with shaking. The reaction was stopped with 100 µL of _1_N H_2_SO_4_, and the absorbance was measured at 450 nm with a Clariostar (BMG Labtech, Ortenberg, Germany) microplate reader. OD values of blank wells, without brain homogenates loaded, were subtracted from OD values of each analyzed sample. Thresholds were calculated as the average of three repeated ELISAs on four spinal cords from young, asymptomatic 1- to 2-month-old M83^+/+^ mice, plus three times the standard deviation (SD) for each ELISA.

### 2.7. Immunohistochemistry

After dissection, brain samples were fixed in 4% paraformaldehyde, then paraffin-embedded and cut into serial 5 µm coronal brain sections. Slides were deparaffinized and rehydrated. Sections were treated with a citrate solution with a pH of 6.2 (ref C9999 Sigma Aldrich, Saint Louis, MO, USA) with microwave heating, then with 10 µg/mL of proteinase K (ref. EU0090-B Euromedex, Souffelweyersheim, France) in water for 20 min at RT. For the detection of aS phosphorylated at serine 129 (pSer129-aS) using the EP1536Y antibody, sections were additionally treated with a 4 M solution of guanidinium thiocyanate (ref. EU0046 Euromedex) for 20 min and washed before the PK treatment. Endogenous peroxidase was suppressed with 30% hydrogen peroxide dissolved in water for 5 min. Non-specific staining was blocked with a blocking reagent (ref 11096176001 Roche, Meylan, France) diluted in maleic acid buffer for 30 min at RT. Primary antibodies were applied (EP1536Y ref. ab51253 Abcam, at dilution 1:350; D37A6 ref. 4179 Cell Signaling at dilution 1:200) in TBS-T and the slides were incubated overnight at 4 °C. After washing with TBS-T, slides were again blocked with a blocking reagent (ref. 11096176001; Roche, Meylan, France) diluted in maleic acid buffer for 30 min at RT. Sections were treated with anti-rabbit IgG HRP conjugate or anti-mouse IgG HRP conjugate (ref. 4010-04 or 1010-04 CliniSciences, Nanterre, France) diluted at 1:250 in TBS-T, for 1 h or 30 min, respectively, at RT. After being washed in TBS-T and TBS, the sections were incubated with an ImmPACT DAB peroxidase (HRP) substrate (ref. SK-4105 Vector Laboratories, Newark, CA, USA) for 8 min at RT and then washed in water. The slides were then counterstained with hematoxylin. Brain images were captured using an Olympus BX51 microscope with ×40 objective. This microscope is equipped with a Lumenera INFINITY3 camera, and Lumenera software (Infinity analyze version 6.5.5) was used.

### 2.8. Statistical Analysis

Data are presented as means +/− standard deviations. Data for Kaplan–Meier curves were analyzed using GraphPad Prism software 6.07. Kaplan–Meier curves and differences in survival periods of M83^+/+^ mice inoculated with PMCA products or control samples were analyzed using a log-rank Mantel–Cox test. A variance analysis was used for the boxplot of the ELISA data obtained during the ageing of M83^+/+^ mice. For each experiment, the mean measures of the OD values obtained using an ELISA test for the mice from 10 days old to 10 months old were compared. For the Syn303/D37A6 ELISA analysis, as the homoscedasticity assumption was verified by the Levene test between age groups, the t-test was performed for the comparisons. However, for the ELISA 4D6/C-20R experiment, the Wilcoxon rank test was performed. The difference was significant when * *p* < 0.05, ** *p* < 0.01 and *** *p* < 0.001.

## 3. Results

We previously reported that the PMCA process of the sonication–incubation of brain homogenate from M83 transgenic mice expressing the human A53T protein specifically induced the aggregation of aS in vitro (Appendix A), and that PMCA also induced the aggregation of aS in C57Bl/6 mouse brain homogenate after being seeded with brain homogenate from sick M83 mice [18].

### 3.1. PMCA Induces the Aggregation of Murine aS in M83 Mouse Brains

To characterize the aggregation of aS in the PMCA products, we analyzed its PK resistance using a panel of different antibodies. In M83^+/−^-PMCA samples, aS showed resistance to proteinase K (PK) digestion up to 100 µg/mL using a Western blot detection with the C-20R antibody, and up to at least 1 mg/mL after the detection with clone 42 (90–140 and 91–96 sequences, respectively) (Figure 1A, left column). We obtained a PK-resistant truncated aS^res^ fragment with a molecular weight < 10 kDa using the antibody clone 42. Using antibodies against either human or murine aS (4B12 (103–108 sequence) and D37A6 (103–110 sequence), respectively), similar aS^res^ fragments were observed, showing that both human and murine aSs were resistant to proteinase K in the 103–110 aS region. Conversely, the C-terminal end of the protein was undetected using the antibodies MJFR1 (118–123 human sequence) or 4D6 (epitope around serine 129) (Appendix A). No aS^res^ was observed from the unseeded C57Bl/6-PMCA samples with C-20R, clone 42 or D37A6 antibodies, even at the lowest 1 µg/mL PK concentration tested (Figure 1A, right column). However, when the C57Bl/6 brain was seeded with a minute quantity of the brain from sick M83^+/+^ mice, murine aS showed similar PK resistance after two PMCA rounds, while limited proteinase K resistance was detectable during the first round (Figure 1B).

We then analyzed the C57Bl/6-PMCA (M83^+/+^-seeded) samples (without PK treatment) using ELISAs. Using a first sandwich ELISA format with the 4D6 capture antibody prior to detection with the C-20R antibody [18], we detected immunoreactivity only in the samples of C57Bl/6 seeded with sick M83^+/+^ mouse brain homogenate; this signal was higher after a second PMCA round (Figure 1C). Similar results were obtained using a second sandwich ELISA with a Syn303 capture antibody and D37A6 for detection. All together, these results provided approaches to further characterize the aggregation of aS occurring in vivo in the M83 transgenic mouse model, including the aS resistance to PK digestion and the analysis of murine aS, neither of which has yet been thoroughly examined.

### 3.2. Alpha-Synuclein from a Sick M83 Brain Is Highly Resistant to Proteinase K

Having shown these aS molecular features after in vitro aggregation using PMCA, we assessed the extent to which they were also found in vivo in M83 mice. We studied a panel of eight old M83^+/+^ mice (aged between 14 and 21 months) sacrificed either in the absence of clinical signs (healthy (H); *n* = 3) or with the typical clinical signs of paralysis (sick (S); *n* = 5). The Western blot detection of serine 129 phosphorylated aS (pSer129-aS) in brain homogenates with the EP1536Y antibody showed a higher signal for monomeric aS and high molecular weight bands only in the five mice with clinical signs of paralysis (Figure 2A, upper panel). After PK digestion, aS^res^ was not detected with the EP1536Y antibody (Appendix A). PSer129-aS accumulation was confirmed in the five sick M83^+/+^ mice (Figure 2B) using an ELISA after capture with the Syn303 antibody and EP1536Y detection [16].

Undigested samples from old M83^+/+^ mice contained two or three truncated aS forms revealed using the clone 42 antibody (Figure 2A, second panel). The lowest of these bands, at < 10 kDa, was detected only in the five sick M83^+/+^ mice. It has already been described as disease-specific in M83 mice [32] and in another human A53T aS expressing transgenic mouse model [45]. When we analyzed aS PK resistance using clone 42 antibody detection at the highest (1 mg/mL) PK concentration, aS^res^ was only detected in the five sick mice (Figure 2A, third panel), indicating that pSer129-aS accumulation and high aS PK-resistance are closely associated. AS^res^ was also detected only in the sick mice using 4B12 and D37A6 antibodies, but only after an increased exposure time and, for two of the five mice, with antibody D37A6 (Appendix A).

The ELISAs revealed increased immunoreactivity levels in the spinal cords of the five sick M83^+/+^ mice, but not in the three old but still apparently healthy M83^+/+^ mice, using 4D6/C-20R and Syn303/D37A6 ELISAs (Figure 2B); this was also observed using a Syn303/EP1536Y ELISA detecting pSer129-aS [16].

### 3.3. Neuro-Anatomical Distribution of Disease-Associated aS in M83 and APPxM83 Mice

To go further in the characterization of the neuropathology in mice, we first examined in more detail the cerebral distribution of disease-associated aS in old and sick M83^+/+^ mice. Using a Western blot analysis, aS was identified in all the seven analyzed brain regions and in the spinal cord with the clone 42, 4B12 and D37A6 antibodies (Figure 3A). Lower levels of murine aS were detected with the D37A6 antibody in the midbrain/brainstem, which was also seen with the clone 42 and C20-R antibodies in wild-type mice (Appendix A) and previously in young M83 and APPxM83 mice [34]. We then examined aS protease resistance in either sick M83 or old APPxM83 mice, in comparison with the pSer129-aS distribution. AS^res^ was detected only in the midbrain, brainstem and spinal cord with both the clone 42 and 4B12 antibodies. A faint signal was also detected with the D37A6 antibody, but this was only revealed after an enhanced exposure time and an increased protein deposition for this antibody (Figure 3A and Appendix A), which is consistent with the detection of pSer129-aS (upper panel) and as previously described in sick M83 mice [9,16]. The distribution of immunoreactivity levels in the different brain regions examined using either the 4D6/C-20R ELISA—previously shown to specifically recognize aS aggregates in M83-PMCA samples [18]—or the Syn303/EP1536Y ELISA that recognizes pSer129-aS [16], were also similar, with much higher levels in the midbrain/brainstem than in the cerebral cortex/striatum (Figure 3B).

Immunoreactivity levels were also found to be higher with the Syn303/D37A6 ELISA in the midbrain/brainstem than in the cerebral cortex/striatum or in B6C3H wild-type mice (Appendix A). This suggests a recruitment of murine aS in the brain lesions of M83 mice, as previously suggested [8,9]. Immunohistochemistry revealed that the pSer129-aS pathology—presenting as cytoplasmic and neuritic deposits—appeared strong in the midbrain/brainstem (Figure 4A). Indeed, cells with cytoplasmic and neuritic staining using the D37A6 antibody were observed in these brain regions (Figure 4B), although this staining was less intense than the abundant pSer129-aS staining (Figure 4A).

To better understand the relationship between pSer129-aS, aS^res^ and ELISA-detected aS aggregates, we built on the legacy of our recent observations in M83^+/+^ mice crossed with APPPS1-21 mice expressing mutated forms of human β-amyloid (APP) and presenilin 1(PS1) proteins [35]. The brain lesions observed were predominantly found in the cerebral cortex, hippocampus and striatum in the hybrid (APP×M83) mice sacrificed, without clinical signs, at the age of 22 months [34]. This distribution of brain lesions thus appeared quite differently from that found in the sick M83 mice. The accumulation of pSer129-aS was readily identified in these brain regions using a Western blot analysis with additional bands having a higher molecular weight (Figure 3C, upper panel) (*n* = 2). After 1 mg/mL PK digestion, aS^res^ was clearly detected in the cerebral cortex, hippocampus and striatum of the APPxM83 mice with clone 42, 4B12 and D37A6 antibodies. This result contrasts with that of the sick M83^+/+^ mice, where aS^res^ was found only in the midbrain/brainstem. In comparison, the APPPS1-21 control mice showed no PSer129-aS accumulation nor aS^res^ (Appendix A). The ELISA analyses of selected brain regions showed high immunoreactivity levels with a Syn303/D37A6 ELISA in the cerebral cortex and striatum. These levels were similar to those observed with Syn303/EP1536Y or 4D6/C-20R ELISAs (Figure 3D). Immunohistochemical analyses also showed cytoplasmic and neuritic labelling with the D37A6 antibody in the cerebral cortex, hippocampus and striatum (Figure 4B), which are the brain regions in which pSer129-aS accumulates most (Figure 4A) [34]. Overall, this showed a strong correlation between D37A6 detections of aS^res^ using Western blot analyses and of aS accumulation using ELISA or IHC tests, suggesting increased murine aS aggregations in distinct brain regions among the APP×M83 mice.

### 3.4. Intrastriatal Inoculation of PMCA Samples Containing Only Murine Aggregated aS Accelerates the M83 Disease

To further investigate the potential of murine aS aggregation, we decided to determine the pathogenicity of PMCA-generated aS aggregates obtained after seeding a C57Bl/6 mouse brain substrate with sick M83^+/+^ mouse brain homogenate. We performed a comparison with the previously reported acceleration of the M83 disease after intrastriatal inoculations of M83-PMCA samples [18]. Two-month-old M83^+/+^ mice were inoculated in the striatum with C57Bl/6-PMCA (M83^+/+^-seeded, dilution 10^−5^, second PMCA round) samples (Experiment 3 in Table 1). All the mice showed the onset of paralysis before the age of ten months, except one (Figure 5A and Table 1).

Conversely, none of the mice inoculated with a C57Bl/6-KO sample spiked with sick M83^+/+^ mouse brain homogenate (dilution 10^−5^) (Experiment 4) or with a C57Bl/6-KO-PMCA sample (Experiment 5) had developed the disease by the age of 10 months. These results demonstrate that the intracerebral inoculation of C57Bl/6-PMCA samples obtained after seeding with sick M83 brains accelerates the M83 disease. This finding is similar to that previously described with M83-PMCA samples [18] and confirmed here (Experiment 2), and also with brain extracts from sick M83 mice [7,9,16,34]. Thus, PMCA products, essentially containing mouse aS, that has been misfolded after seeding with brains from sick M83^+/+^ mice, are highly pathogenic for M83 mice.

We then characterized both the mice inoculated with M83-PMCA or C57Bl/6-PMCA (M83^+/+^-seeded) samples by studying the distributions of ELISA immunoreactivity levels in the different brain regions and spinal cord using a 4D6/C-20R ELISA to identify aS aggregates, and a Syn303/EP1536Y ELISA to recognize pSer129-aS. Whatever the inoculum, the immunoreactivity levels obtained from both the pSer129-aS and 4D6/C-20R ELISAs were mainly identified in the midbrain/brainstem and spinal cord (Figure 5B). Using the Syn303/D37A6 ELISA, a higher immunoreactivity level was also observed in the midbrain/brainstem and spinal cord; this again suggests the recruitment of murine aS in lesioned brain regions.

As previously described in old and sick M83 mice, aS^res^ bands were consistently detected by the Western blot analysis in the midbrain/brainstem and spinal cord after the 1 mg/mL PK digestion with the clone 42 antibody (~ 7–9 kDa), but only at minimal levels with the D37A6 antibody (~9 kDa), and only after an increased protein deposit (Figure 6A,C and Appendix A). Using immunohistochemistry, pSer129-aS labelling revealed cytoplasmic and neuritic deposits in the midbrain/brainstem and deep cerebellar nuclei in the mice inoculated with both the C57Bl/6-PMCA (M83^+/+^-seeded) and M83-PMCA samples. Some pSer129-aS accumulation was also detected in the cerebral cortex and striatum of the mice inoculated with the C57Bl/6-PMCA (M83^+/+^-seeded) sample (Figure 6D). No specific labelling could be observed with either the D37A6 or EP1536Y antibodies in an additional group of mice that were inoculated with a M83^+/+^-standing sample and sacrificed at the same ages as those developing paralysis following inoculation with the M83^+/+^-PMCA sample (Figure 6B). Thus, despite the difference in the substrate origin (M83 versus C57Bl/6 brain), PMCA-generated aS aggregates similarly accelerated the M83 disease and induced a similar disease phenotype.

### 3.5. Early and Widespread aS Aggregates Can Be Detected in M83 Mice

We previously showed that aS aggregates could be detected at the terminal stage of the disease. Here, we further investigated the spatiotemporal detection of aS aggregates using ELISAs on the brains of asymptomatic M83^+/+^ mice at the early stages. For this, we used a cohort of M83^+/+^ mice that were sacrificed at different timepoints from 10 days to 10 months old (*n* = 6 per age) (Figure 7A). The results showed an early rise in 4D6/C-20R immunoreactivity levels, which significantly increased between 10 days and 10 months in all the brain regions and in the spinal cord (*p* = 2.17 × 10^−3^ and *p* = 7.96 × 10^−3^, respectively) (Figure 7B). The 10-month-old outlier mouse was excluded from the statistical analysis, visualized in the figure. Surprisingly, the highest immunoreactivity levels were found in the cerebral cortex and cerebellum, then in the striatum, whereas lower immunoreactivity levels were found in the hippocampus, midbrain, brainstem and olfactory bulbs; immunoreactivity was the lowest in the spinal cords. The Syn303/D37A6 immunoreactivity levels (Figure 7C) were lower, but also significantly increased at the age of 10 months in the cerebral cortex, striatum and cerebellum. In contrast, no noticeable immunoreactivity was observed in any brain region or in the spinal cord of the B6C3H mice sacrificed at the age of 7 months (Appendix A) with both 4D6/C-20R and Syn303/D37A6 ELISAs.

The outlier values were observed for the only symptomatic 10-month-old mouse in this experiment, which had a high 4D6/C-20R immunoreactivity level in the midbrain, brainstem and spinal cord, where it also had pSer129-aS and aS^res^ (Figure 7D,E). This symptomatic mouse had a slightly increased immunoreactivity level in the midbrain/brainstem and spinal cord as revealed using a Syn303/D37A6 ELISA. This finding had previously been observed in other sick M83^+/+^ mice (Figure 2B and Figure 3B). Conversely, pSer129-aS was not detected in either of these brain regions or the spinal cord of the other five M83^+/+^ mice sacrificed at the age of 10 months, again indicating the late and consistent detection of both pSer129-aS and aS^res^ (Figure 7D,E).

## 4. Discussion

We previously reported the in vitro amplification of the aS misfolding of a normal mouse brain substrate using seeds from the M83 transgenic mouse model overexpressing the A53T mutated human aS and the well-established PMCA method developed for prions [18]. To our knowledge, this is the first report of aS-PMCA using a brain-derived substrate; previous studies have used recombinant aS protein as a substrate [10,46,47,48,49]. Proteinase K-resistant aS^res^ fragments were readily detected in the M83-PMCA samples obtained using the PMCA of M83 mouse brains through antibodies against the 91–110 region of the protein, such as the clone 42 antibody (91–96 sequence). Both human and murine-specific 4B12 and D37A6 antibodies, respectively, (103–110 and 103–108 sequences) also readily identified aS^res^, indicating that PMCA induced the aggregation of the endogenous murine aS in addition to human A53T mutant aS. Murine aS^res^ was also identified after the PMCA of a C57Bl/6 mouse brain substrate, but only when the reaction was seeded with the brains from sick M83 mice. The aggregation of murine aS is able to be revealed by observing its PK resistance and using ELISA detection with the D37A6 antibody after the Syn303 capture antibody. Most importantly, this study clearly shows that such C57Bl/6-PMCA samples are pathogenic for M83 mice after intracerebral inoculations, as previously described using M83-PMCA samples [18]. Phenotypical features of the disease were similar to those observed after inoculations of M83-PMCA samples [18], sick M83 mouse brains, or recombinant aS preformed fibrils [9] with regard to both aS^res^ and pSer129-aS detection in the midbrain/brainstem and spinal cord using ELISA and/or immunohistochemistry tests. It may be noted that no evidence of a major increase of the involvement of murine aS was found in the sick M83^+/+^ mice inoculated with C57Bl/6-PMCA (M83^+/+^-seeded) samples. However, this finding clearly shows that M83 aS misfolding is transmitted to murine aS during the in vitro process, leading to its pathogenicity at least for M83 mice. This is similar to the way that prions cross transmission barriers, which is a crucial phenomenon in prion disease transmission that can be reproduced to some extent in vitro during PMCA experiments [50].

The proteinase K resistance of aS in M83 mice was assessed with the 4B12/clone 42 antibodies and found to be similar in old and spontaneously sick mice or in those showing an early onset of paralysis after intracerebral inoculations of brain homogenate from sick M83^+/+^ mice or of PMCA samples containing aggregated aS. Although aS^res^ was undetected with antibodies against the C-terminal aS end—such as the 4D6 or MJFR1 antibodies against the 124–134 and 118–123 regions, respectively (Appendix A)—it was readily detected by the 4B12/clone 42 antibodies, specifically in the midbrain/brainstem or spinal cord, which contain the highest pSer129-aS levels in sick M83 mice. While pSer129-aS is often used as a pathological biomarker of synucleinopathies [51,52], high resistance to PK digestion is the biochemical hallmark of the prion protein (PrP) in prion diseases. In most of them, however, the N-terminal end of this protein is removed. Overall, these data are consistent with those previously reported in M83 mice [32]. A recent review emphasized the critical role of aS proteolytic processing in vivo, which can be augmented by dysfunctional proteostasis and dramatically potentiates the propensity of aS to pathologically misfold into uniquely toxic fibrils with modulated prion-like seeding activity [53]. Interestingly, PK resistance was also found in aS aggregates produced in vitro, including after seeding a C57Bl/6 mouse brain substrate with a sick M83 mouse brain homogenate. While this was associated with the finding of SDS-resistant dimers in the Western blot analysis, it indicates that protease resistance does not require the phosphorylation of the protein. Although, at minimal levels (after increased exposure times or increased protein deposits), murine aS^res^ was detected using the D37A6 antibody in the brain of sick M83^+/+^ mice both during ageing and after intracerebral inoculations of PMCA products. In addition to a seemingly limited aggregation of murine aS compared with the human A53T mutated aS of M83 mice, the limited detection from the Western blot analysis after the PK digestion may be due to a partial cleavage of this epitope after the PK digestion, located at the C-terminal end of aS^res^, together with the limited sensitivity of the Western blot analysis [32,33,54]. The association of murine aS with brain lesions is definitely corroborated by both the ELISA immunoreactivity levels and immunohistochemical labelling in the midbrain/brainstem regions using D37A6 antibody detection. A recruitment of murine aS within aS aggregates has previously been reported in M83 mice, with the co-labelling of human and murine aS within intracytoplasmic inclusions [8], which is also supported by our immunohistochemical observations using the D37A6 antibody. In the study by Luk et al., Western blot analyses failed to reveal murine aS in formic acid fractions from sick M83 mouse brains, unlike pSer129-aS. However, we managed to detect minute amounts of aS^res^ using a Western blot analysis in denaturing conditions with the D37A6 antibody, which was previously used to analyze the specific contribution of murine aS in the aggregation process [25,55]. Interestingly, it has previously been shown that the absence of murine aS promotes the specific aggregation of human aS in a transgenic mouse model expressing wild-type human aS, and that murine aS preferentially interacts with aggregated human aS, attenuating seeding and spreading [25]. We can speculate on whether the lower expression of murine aS in the midbrain/brainstem and spinal cord [34] could contribute to the greater presence of lesions in these neuro-anatomical regions of M83 mice. Overall, the current study revealed a qualitative correlation between the findings of pSer129 aS in vivo and aS^res^, which was only identified after the in vitro PK digestion. A limitation of our study is thus that it does not clarify the precise relationship between these aS forms in the pathogenesis of the disease in mice. Since studies in M83 mice require their observation up to the clinical stage, the time at which aS^res^ and PSer129-aS appear during the incubation of the disease remains unknown. Recent studies have shown that truncated aS species detected in vivo using immunohistochemical analyses with specific antibodies could appear at early stages after the intra-muscular inoculation of fibrillar aS in M83 mice [33]. In humans with PD, one study has shown the role of asparagine endopeptidase (AEP) in aS cleavage, triggering its aggregation and escalating its neurotoxicity [56]. In this work, some in vitro studies did not indicate interference between aS phosphorylation and AEP aS cleavage.

We recently observed in APP×M83 mice (obtained by crossing M83 mice with APPPS1-21 mice overexpressing human mutated beta-amyloid and presenilin 1 [35]) the accumulation of pSer129-aS mainly in the cerebral cortex/hippocampus and striatum during ageing, which is very different from sick M83 mice [34]. Accordingly, in this study, aS^res^ was readily detected in these brain regions of old APPxM83 mice using the clone 42 antibody. This again illustrates the close relationship between the presence of pSer129-aS and PK-resistant aS species. An increased aggregation of murine aS was revealed in these same brain regions using a Western blot detection of aS^res^ with the D37A6 antibody. Indeed, the signals were higher than those observed with the human-specific 4B12 antibody, associated with increased ELISA immunoreactivity levels using the Syn303-D37A6 ELISA and with cytoplasmic and neuritic labelling using immunohistochemical testing with the D37A6 antibody. However, the intrastriatal inoculation of an APPxM83 cerebral cortex sample containing abundant pSer129-aS in M83^+/+^ mice did not show any evidence of changes in the disease phenotype, compared with that previously described in M83^+/+^ mice [34]. An increased aggregation of murine aS could be fostered by a higher expression of murine aS in these brain regions [34]. In comparison with M83 mice, it could also be an effect associated with beta-amyloid/presenilin 1 expression in these mice, since these brain regions show the earliest signs of amyloid plaque accumulation in APPPS1-21 mice [35]. In another beta-amyloid/presenilin 1 transgenic mouse line (5×FAD) expressing murine but not human aS, the amyloid plaque environment dramatically facilitated or promoted the aggregation and spreading of murine aS after intracerebral inoculations of murine aS recombinant fibrils or of human aS derived from Lewy body pathology [57]. The aggregation of aS via heterogeneous primary nucleation triggered by soluble aggregates containing amyloid-β was evidenced recently in vitro [58]. The availability of both APP×M83 and M83 mouse models, with strikingly distinct neuropathological features, may represent a unique opportunity for further studies of translational relevance for human diseases, especially in the context of the increased awareness of the importance of co-pathologies [3]. The M83 mouse model is already used to assess potential therapies for humans by investigating neuropathologies associated with motor deficits [59], or earlier behavioral and cognitive deficits of these mice [60].

PSer129-aS is known to be a late biomarker of the disease in M83 mice [61]. It was recently suggested that it occurs after protein aggregation and inhibits seeded fibril formation and toxicity [62]. To look for earlier aS biochemical modifications in this model, we examined a cohort of M83^+/+^ mice ranging from 10 days to 10 months old to detect aS in different brain regions using ELISAs. Only one out of six mice sacrificed at the age of 10 months had an accumulation of pSer129-aS in the midbrain/brainstem and spinal cord, which is generally consistent with the minimal age at which the onset of paralysis in M83 mice was observed in our laboratory [7] and with previously reported data regarding the onset and distribution of the pathology in M83 mice [63]. However, even in the absence of any pSer129-aS detected at earlier ages, 4D6/C-20R and Syn303/D37A6 ELISA immunoreactivity levels (used to detect aS aggregates) were found to increase progressively with age up to the maximal age of 10 months examined. Surprisingly, this increase was much more limited in the midbrain/brainstem and spinal cord than in the cerebral cortex/cerebellum, where it was found with a 4D6/C-20R ELISA. This was unexpected because these two brain regions show limited lesions at the terminal stage of the disease in M83 mice. However, a previous study in M83 mice revealed the early detection of detergent-soluble aS oligomeric forms in brain regions devoid of aS inclusions, such as the *substantia nigra* and olfactory bulbs [64]. The significance of early aS aggregates in M83 mice remains to be established. Besides the late occurrence of paralysis in this mouse model, several studies have reported their significant behavioral or motor impairments as early as 1–2 months of age [65,66]. Moreover, recent studies in the G2-3 mouse line (comparable to M83 mice) have revealed the unique role of the A53T mutation in the human protein in early memory and synaptic deficits, independently of neurodegenerative changes [67,68]. By reducing the aggregation of aS in the visual cortex and hippocampus, rifampicin treatment could mitigate early cognitive dysfunction preceding the late motor clinical signs of G2-3 mice [69]. A recent study in humans with PD showed that the aS oligomer burden was significantly greater in the neocortex, while the Lewy-related pathology was greater in vulnerable subcortical regions—including the brainstem—suggesting that aS oligomers may be widely distributed early in the process of PD [70].

Here, we describe the compared aS aggregation features in two mouse models expressing the A53T human protein, either alone (M83) or together with human mutated β-amyloid/presenilin 1 (APP×M83). Notably, we report novel findings regarding aS protease resistance or ELISA assessments of aS aggregation in these two distinct models, in comparison with previously published data focused on Pser129-aS. However, we did not address the functional consequences associated with the expression of these transgenes and following aS aggregate inoculations. It may involve an assessment of neurodegenerative changes or synaptic and mitochondrial deficits. It is noteworthy that the M83 mouse model does not show major lesions of the nigro-striatal pathway [6]. For this very reason, it could be used to dissect the molecular mechanisms linking the two cardinal features of PD, dopaminergic cell death and aS aggregation [71]. Increasing dopamine in M83 mice triggered the loss of terminals that precedes nigral cell death, suggesting synaptic involvement in the disease. The development of a well-characterized mouse model such as the one described here in APP×M83 offers a unique opportunity for further studies, notably in relation with the striatal involvement. Also, the high levels of lesions found in the cortico–hippocampal regions may help to tackle their functional consequences, with potential relevance to Lewy bodies dementia or PD with dementia.

Our study thus provides novel knowledge with which to decipher the molecular mechanisms involved in the pathophysiology of synucleinopathy in the M83 model, which is frequently used to assess a variety of therapeutic strategies to tackle these devastating diseases [72,73,74,75]. It notably illustrates how the specific behavior of murine aS can be monitored by both biochemical and immunohistochemical approaches, questioning its aggregation during the disease. However, our study importantly shows that brain samples expressing only wild-type mouse aS acquire pathogenicity for M83 mice after in vitro misfolding using PMCA. After seeding wild-type mouse brains with the brains of sick M83 mice, features of the disease in M83 mice inoculated with the amplification product, containing murine aS, are similar to that previously described after inoculations of sick M83 mouse brains [9] or of PMCA-aggregated aS from M83 mouse brains [18]. This may open the way for further studies with other PMCA sources of aggregated aS in this mouse model, in which distinct phenotypical features could be revealed when challenged with different aS “strains” [10,16,76,77].

## Figures and Tables

**Figure 1 biomolecules-13-01788-f001:**
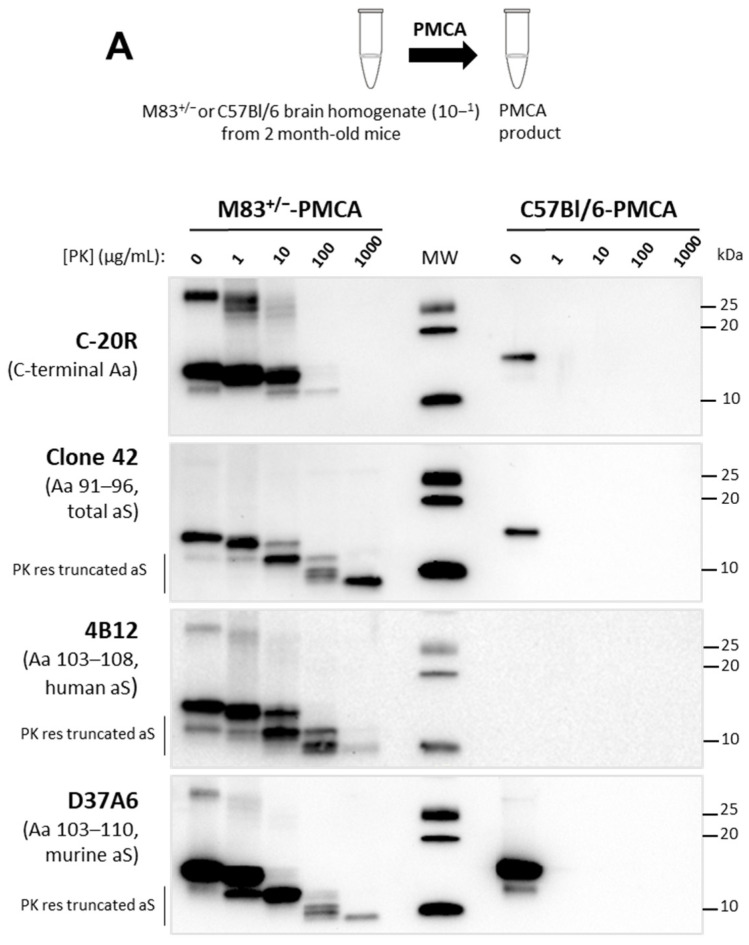
PMCA triggers aggregation and PK resistance in M83 and C57Bl/6 (M83-seeded) mouse brain samples. (**A**) 10% brain homogenates from 2-month-old healthy M83^+/−^ and C57Bl/6 mice were submitted to 144 cycles of PMCA (M83^+/−^-PMCA and C57Bl/6-PMCA samples). Western blot analysis of aS after PK digestion from 1 to 1000 µg/mL final concentration of PK, with C-20R and clone 42 antibodies, as well as with D37A6 and 4B12 antibodies specifically recognizing murine and human aS, respectively. Truncated aS^res^ was identified with antibodies against the central aS region (clone 42, 4B12 and D37A6) in the M83-PMCA sample. Molecular weights are stated on the right (in kDa). (**B**,**C**) Seeding of C57Bl/6 mouse brain substrate with sick M83^+/+^ brain homogenate (dilution 10^−4^) after 1 or 2 rounds (144 cycles each) of PMCA. (**B**) Western blot analysis of aS after PK digestion, with the clone 42 and D37A6 antibodies. Truncated aS^res^ was identified with antibodies against the central aS region (clone 42 and D37A6) only in the C57Bl/6-PMCA sample (M83-seeded, 10^−4^ dilution). (**C**) ELISA immunoreactivity of aS (without PK digestion) for C57Bl/6-PMCA ((M83^+/+^-seeded (10^−4^ dilution) or unseeded) samples using a sandwich ELISA with the 4D6 capture antibody prior to C-20R detection, or with the Syn303 capture antibody prior to D37A6 detection.

**Figure 2 biomolecules-13-01788-f002:**
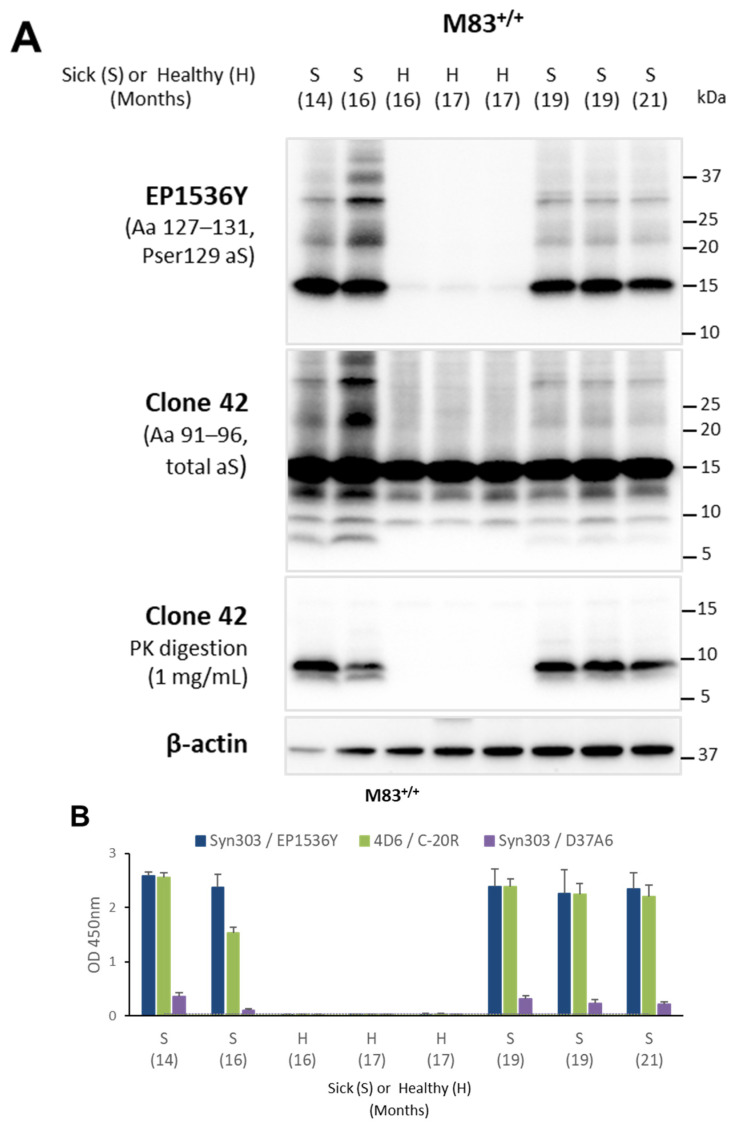
Aggregation and PK resistance in spontaneously sick M83^+/+^ mice. Spinal cord samples from a panel of eight old mice (14–21 months) were analyzed using Western blot or ELISA. (**A**) Western blot analysis of serine 129 phosphorylated aS (pSer129-aS) (antibody EP1536Y) and total aS (antibody clone 42) from brain samples without or after PK digestion at 1 mg/mL. An antibody against β-actin was used as a control of protein load. Molecular weights are stated on the right (in kDa). (**B**) Aggregated aS or pSer129-aS levels were identified using sandwich ELISAs with 4D6/C-20R, Syn303/EP1536Y [16] or Syn303/D37A6 antibodies, respectively. Thresholds were calculated as the average of three repeated ELISA tests on four negative spinal cords from young, asymptomatic M83 mice of 1 and 2 months of age, plus three times the SD for each ELISA test. Three replicate ELISAs were performed for each sample.

**Figure 3 biomolecules-13-01788-f003:**
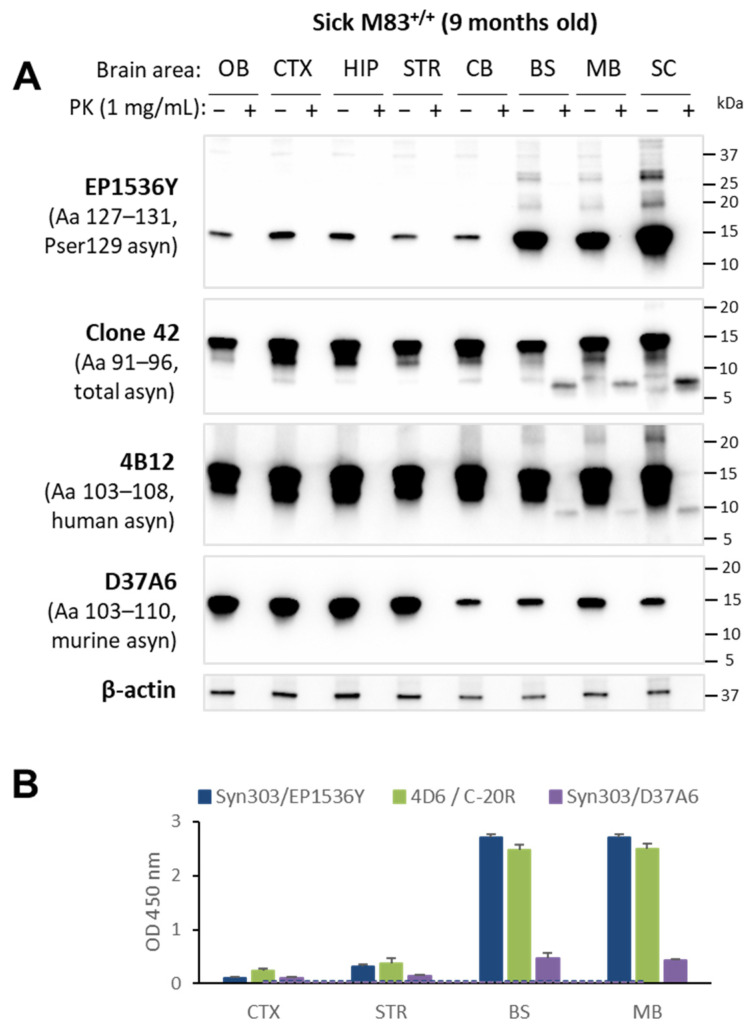
Neuroanatomical distribution of disease-associated human and murine aS in sick M83^+/+^ mice and in old APPxM83 mice. M83^+/+^ (**A**) or APPxM83 (**C**) brain homogenates were incubated without (−) or with (+) PK (1 mg/mL at 37 °C for 30 min). Samples were analyzed by a Western blot using antibodies EP1536Y against pSer129-aS, clone 42, 4B12 (human-specific) and D37A6 (murine-specific) or an antibody against β-actin as a control of protein loads. Bars to the right of each Western blot panel indicate the molecular weight markers (kDa). M83^+/+^ (**B**) or APPxM83 (**D**) distribution of aS detected using ELISA with Syn303/EP1536Y for pSer129-aS, 4D6/C-20R for aggregated aS and Syn303/D37A6 for murine aS. Results were obtained with 20 µg of proteins. The Western blots for M83 and APPxM83 mice are representative of the results obtained for three mice and two mice, respectively. Thresholds were calculated as the average of three repeated ELISA tests on four negative spinal cords from young (1- or 2-month-old), asymptomatic M83 mice, plus three times the SD for each ELISA test. Three replicate ELISAs were performed for each sample. OB: olfactory bulb; CTX: cerebral cortex; HIP: hippocampus; STR: striatum; CB: cerebellum; BS: brainstem; MB: midbrain and SC: spinal cord.

**Figure 4 biomolecules-13-01788-f004:**
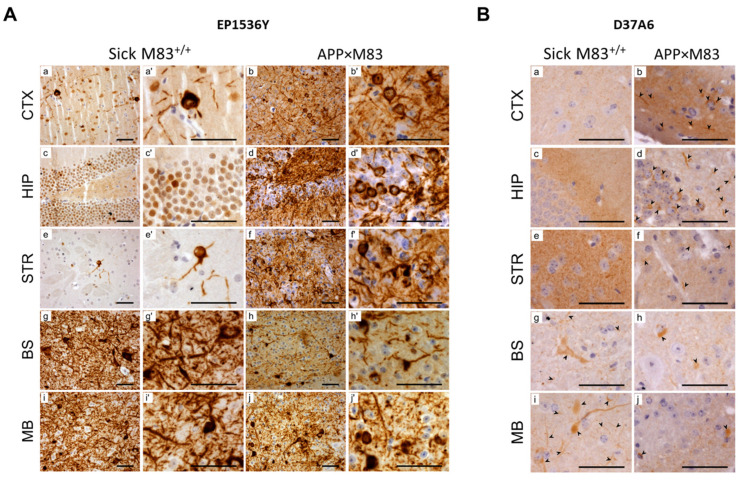
Neuroanatomical distribution of aS analyzed using immunohistochemistry in sick M83^+/+^ mice and in old APP×M83 mice co-expressing mutated human beta-amyloid/presenilin proteins. This shows a representative immunohistochemistry of sick M83^+/+^ mice versus 22-month-old APP×M83 mice in the cerebral cortex (CTX), hippocampus (HIP), striatum (STR), brainstem (BS) and midbrain (MB). (**A**) pSer129-aS detected with the EP1536Y antibody: Representative areas of the different pSer129-aS staining are shown at two different magnifications a, b, c, d, e, f, g, h, i and j and ×3 in a’, b’, c’, d’, e’ and f’, g’, h’,i’ and j’. (**B**) murine aS detected with the D37A6 antibody. Scale bars: 50 µm.

**Figure 5 biomolecules-13-01788-f005:**
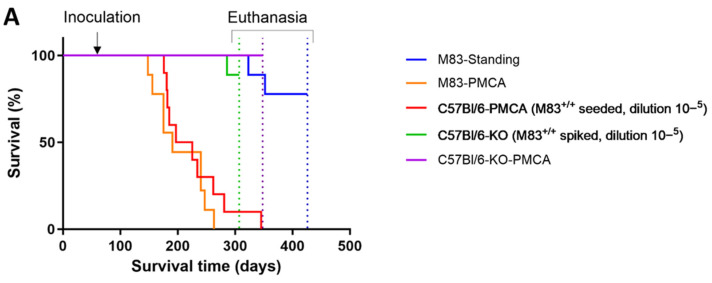
M83^+/+^ mice inoculated with C57Bl/6-PMCA (M83^+/+^-seeded) samples show accelerated progression of the disease. (**A**) Kaplan–Meier curves of the survival of M83^+/+^ mice inoculated at the age of 2 months with M83-PMCA (*n* = 9), C57Bl/6-PMCA (sick M83^+/+^-seeded, dilution 10^−5^ (*n* = 10) in comparison with M83-standing samples (*n* = 9) and C57Bl/6-KO brains spiked with sick M83^+/+^ brain (10^−5^ dilution) (*n* = 9) or not spiked but with PMCA (*n* = 10). * One mouse died of intercurrent disease in the group of mice inoculated with the C57Bl/6-KO sample spiked with sick M83^+/+^ brain. (**B**) Neuroanatomical distributions of aS immunoreactivity levels using ELISA with Syn303/EP1536Y for pSer129-aS, 4D6/C-20R for aggregated aS and Syn303/D37A6 for murine aS. Each symbol (circles and triangles) represents a mouse, and the triangles are the mice whose brains were tested for PK resistance in Figure 6A,C. OB: olfactory bulb; CTX: cerebral cortex; HIP: hippocampus; STR: striatum; CB: cerebellum; MB: midbrain; BS: brainstem and SC: spinal cord, C: contro-lateral side, I: inoculated side.

**Figure 6 biomolecules-13-01788-f006:**
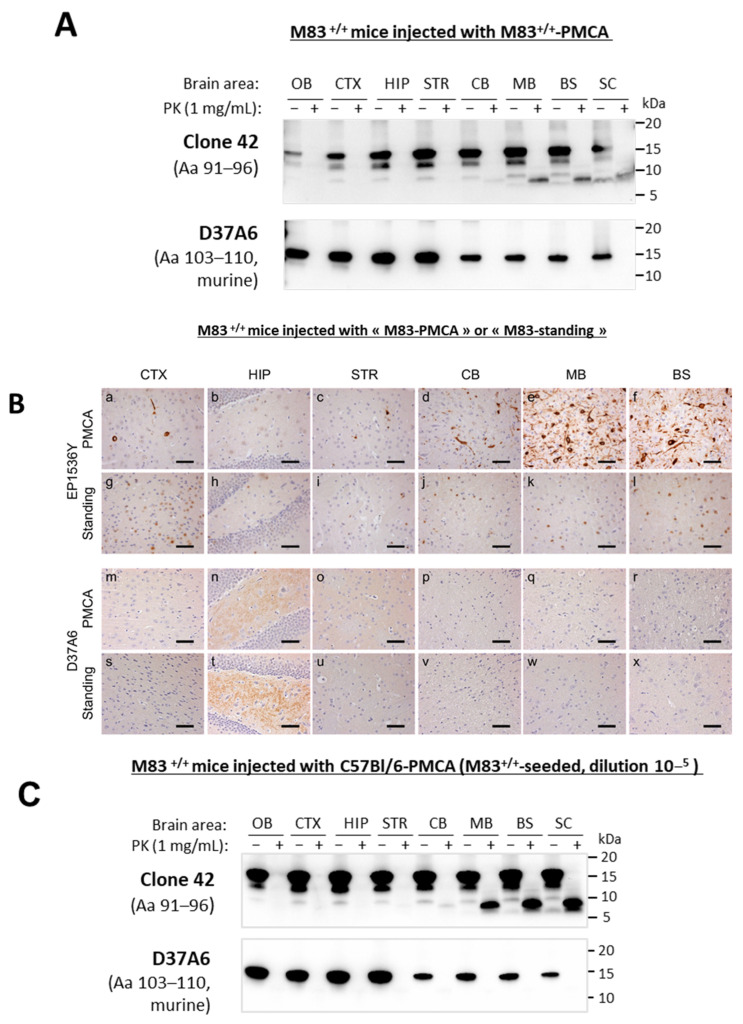
Neuroanatomical distribution of PK-resistant aS (aS^res^) and pSer129-aS deposits in M83^+/+^ mice inoculated with M83-PMCA and C57Bl/6-PMCA (M83^+/+^-seeded) samples. (**A**,**C**)**:** Western blot detection of aS^res^ from different brain areas and spinal cord after digestion with 1 mg/mL of proteinase K at 37 °C for 30 min using clone 42 and D37A6 (murine-specific). Bars to the right of each Western blot panel indicate the molecular weight markers (kDa). Representative results from a sick 8-month-old M83^+/+^ mouse inoculated with a M83-PMCA sample (**A**) or a sick 5-month-old M83^+/+^ mouse inoculated with C57Bl/6-PMCA (M83^+/+^-seeded, dilution 10^−5^) sample (**C**). (**B**,**D**): Representative immunohistochemistry of pSer129-aS (EP1536Y) or murine aS (D37A6) detection. (**B**) M83^+/+^ mice inoculated with M83-PMCA or M83-standing samples sacrificed at the same age as when the mouse inoculated with the M83-PMCA sample developed paralysis. (**D**)**:** M83^+/+^ mice inoculated with the C57Bl/6-PMCA (M83^+/+^-seeded, dilution 10^−5^) sample. Scale bars: 50 µm. OB: olfactory bulb, CTX: cerebral cortex; HIP: hippocampus; STR: striatum; CB: cerebellum; MB: midbrain, BS: brainstem and SC: spinal cord.

**Figure 7 biomolecules-13-01788-f007:**
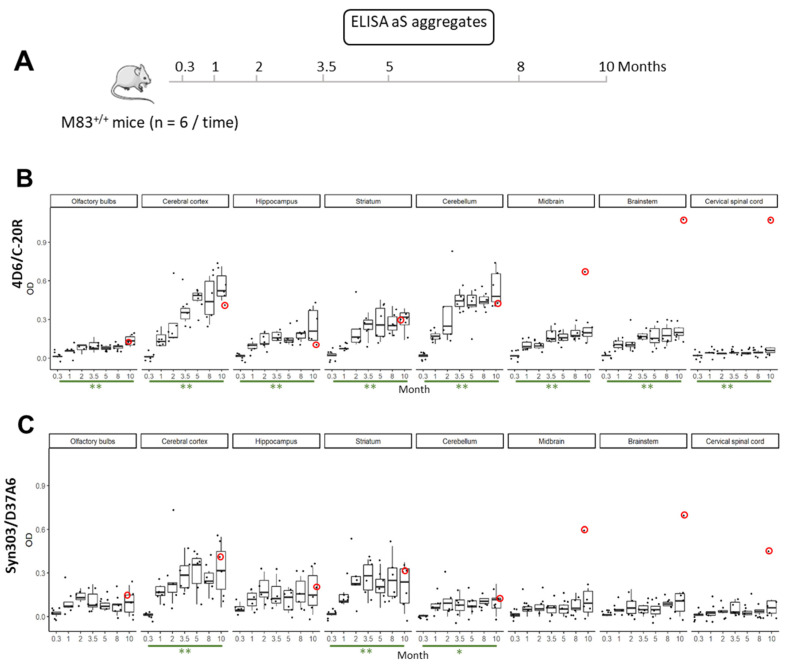
ELISA analysis of aS during ageing in different brain regions of M83^+/+^ mice. (**A**) M83^+/+^ mice (uninoculated) were sacrificed from the age of 10 days to 10 months (*n* = 6 per age) and homogenates were prepared from 7 brain regions and from the spinal cord. Levels of aS immunoreactivity were measured using the 4D6/C-20R ELISA (**B**) or the Syn303/D37A6 ELISA (**C**). The red circles identify the results obtained in the only mouse out of the six mice sacrificed at the age of 10 months that had clinical signs of paralysis. This mouse was excluded from the statistical analysis, visualized in the figure. Boxplot: asterisks indicate the statistical significance of the differences between the aS immunoreactivity levels (* *p* < 0.05, ** *p* < 0.01) in green between 1 and 10 months. (**D**) Immunoreactivity levels detected by the Syn303/EP1536Y ELISA (pSer129-aS) or 4D6/C-20R ELISA (aS aggregates) in the cerebellum (CB), brainstem (BS), midbrain (BS) and spinal cord (SC) of the six mice sacrificed at the age of 10 months. Mouse no. 4 is the only one showing signs of paralysis. (**E**) Western blot detection of aS^res^ using the clone 42 antibody after PK digestion (1 mg/mL, 30 min at 37 °C) in the midbrain, brain stem, cerebellum and spinal cord of mouse no. 4, which showed signs of paralysis at the age of 10 months. Bars to the right of the Western blot indicate the molecular weight markers (kDa).

**Table 1 biomolecules-13-01788-t001:** Incubation periods and attack rates.

Reference	Experiment	Inoculum (Brain Homogenate)	PMCA Round	Number of Sick 83 Mice That Died < 300 Days Old	Age at Death Due to Disease (Days)	Days Post-Inoculation
[1]	M83^+/+^-Standing	Healthy M83^+/+^ 10^−1^	0	0/9		sacrificed at 426
[2]	M83^+/+^-PMCA	Healthy M83^+/+^ 10^−1^	1	9/9	204+/−44	92–207
[3]	C57Bl/6-PMCA (M83^+/+^-seeded, dilution 10^−5^)	Sick M83^+/+^ 10^−5^ in C57Bl/6	2	9/10	227+/−55	120–289
[4]	C57Bl/6 KO spiked with M83^+/+^ 10^−5^	Sick M83^+/+^ 10^−5^ in C57Bl/6 KO	No	0/9		sacrificed at 307
[5]	C57Bl/6 KO-PMCA	C57Bl/6 KO	2	0/10		sacrificed at 348

## Data Availability

The data presented in this study are available on request from the corresponding author.

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
