# Peer review of "Protease-Sensitive and -Resistant Forms of Human and Murine Alpha-Synucleins in Distinct Brain Regions of Transgenic Mice (M83) Expressing the Human Mutated A53T Protein"

_biomolecules, 2023, doi:10.3390/biom13121788_

Round 1
Reviewer 1 Report
Comments and Suggestions for Authors
In my opinion it is a good piece of in-vivo research very focused on some biological topics.
It is difficult to me to understand how these results can lead to signficant contributions in the diagnosis and therapy of the neuordegenerative diseases cited in the Intro.
Maybe the authors can address this point in the discussion and conclusions to improve the impact and potentiality of their research.
Reviewer 2 Report
Comments and Suggestions for Authors
This is an engaging article with robust methodology purposefully designed to study the pathophysiology of synucleinopathy in the M83 model. The manuscript is well-written, and the methods are adequately documented.
My only minor comment refers to the keyword “oligomer”, which should be deleted as no relevant data are presented herein.
Reviewer 3 Report
Comments and Suggestions for Authors
1. Lines 39-40; claim about alpha-synuclein (aS) aggregation is incomplete. The aggregation is comprised of many other components; to which aS is found to be the main proteinaceous part of it. Please change accordingly.
2. Between Fig.1 and 2; were clone-Abs randomly selected or were tested for all but only Clone 42 & D37A6 were able to detect the bands. If others were not able to detect then please include those blots as supplementary data.
3. Fig.2. after PK-digestion how does the blot appears with EP153Y?
4. Are observed truncated forms of aS functionally relevant? They also appear to be showing PK-resistance therefore, whether they hold any pathological relevance is a question that should be answered.
5. pSer129-aS accumulation and high aS PK-resistance shows a correlation but authors should provide more direct evidence of it.
6. Did authors try to figure which part of aS their Abs. E.g., in Fig.3. does these Abs cross-recognize the specified regions. Ideally a library of fragments can be prepared and submitted as supplementary information.
7. Obviously, in continuity it is expected if authors describe whether these Abs are equally efficient in detecting the signal in various brain regions (Fig-3.).
8. Authors should more clearly introduce and articulate the significance of their work in hybrid mice.
9. More clarification is required if there is any age dependent aggregate accumulation observed in the models used in this work and how that is further related to the survival of the animals (Fig.5.)?
10. M83-mice are more relevant to study the disease-onset and pSer129-aS accumulation is rather good sign, however between phosphorylation to aggregation the temporal kinetics is undetermined. For example, in Fig. 5 after the inoculation at what stage the mice start showing pSer129-aS accumulation?
11. In hybrid mice, there is no control study done showing that the observed findings are clearly the result of aS-aggregates? Furthermore, only few Abs were tested, please explain why?
12. Neuroanatomical studies in Fig.6., what about quantifying TH - +ve neurons?
13. As per authors what exactly contribute to the PK-resistance?
14. Can authors include any specificities related to paralysis specificities observed in sick mice?
15. Any other behavioral observation in the mice used?
16. What could authors say about the oligomeric organization of the observed aggregates in their experiments, please include to your discussion.
Comments on the Quality of English LanguageDue to complex sentence structure, the manuscript appears less interesting despite many interesting experiments therefore, it is highly advised to adapt a rather simple style of presentation and develop a clear flow of the arguments.
Round 2
Reviewer 3 Report
Comments and Suggestions for Authors
1. It is advised that authors include their response (comment #7; Obviously, in continuity it is expected.... (Auth-Res: We are not fully sure to understand the question of the reviewer, maybe suggesting different efficiencies of antibodies according............)) to the manuscript.
2. Based on response to comment 8&12, author’s themselves indicate towards the limitations posed by the used model system for the PD-pathology, therefore it is expected to discuss such limitations.
3. In most of the responses, authors try to validate/argue by stating that it is done in their previously published work, this may be showing the continuity of their endeavors, but it is also in contradiction to establish the Novelty of this work.
Comments on the Quality of English LanguageFurther careful review of the use of English language in terms of maintaining an appropriate flow of ideas is needed. Do authors suggest that a ‘native English translator’ is a benchmark to write in a scientific format 😊?
